# In-Context Learning with Retrieval Augmented Encoder-Decoder Language Models

## Abstract

In this paper, we investigate the in-context learning ability of retrieval-augmented encoder-decoder language models. We first conduct a comprehensive analysis of the state-of-the-art Atlas model and identify its limitations in in-context learning, primarily due to a mismatch between pretraining and testing, as well as a restricted context length. To address these issues, we propose Raven, a model that combines retrieval-augmented masked language modeling and prefix language modeling. We further introduce *Fusion-in-Context Learning* to enhance the few-shot performance by enabling the model to leverage more in-context examples without requiring additional training or model modifications. Through extensive experiments, we demonstrate that Raven significantly outperforms Atlas and achieves results comparable to the most advanced language models in certain scenarios, despite having substantially fewer parameters. Our work underscores the potential of retrieval-augmented encoder-decoder language models for in-context learning and encourages further research in this direction.[1]

## 1 Introduction

Recent advancements in natural language processing have been predominantly driven by the development of large language models (LLMs) (Brown et al., 2020; OpenAI, 2022; 2023; Chowdhery et al., 2022; Smith et al., 2022). These models have demonstrated remarkable performance across a wide range of tasks (Qin et al., 2023; Bubeck et al., 2023; Huang & Chang, 2023). One of the key features that enables these models to excel is their ability to perform in-context learning (Dong et al., 2022). By conditioning on given context, LLMs can adapt to new tasks and domains without the need for task-specific fine-tuning. This enables LLMs to perform well on zero-shot or few-shot learning tasks, where only a limited number of examples are available.

While in-context learning has been extensively studied for decoder-only language models like GPT-3 (Brown et al., 2020) and PaLM (Chowdhery et al., 2022), research on encoder-decoder language models, which have shown to learn stronger representations (Devlin et al., 2019; Raffel et al., 2020), remains limited. Notably, Patel et al. (2023) tap into the potential of mT5 (Xue et al., 2021), a multilingual encoder-decoder LM, by iteratively prompting the model to produce long generations with in-context examples. Chung et al. (2022); Longpre et al. (2023) finetune T5 (Raffel et al., 2020) with a large mixture of tasks using instruction tuning (Mishra et al., 2022; Wei et al., 2022; Sanh et al., 2022) to improve model performance and generalization to unseen tasks in both zero-shot and few-shot settings.

On the other hand, LLMs still face challenges such as hallucination and limitations in representing the long-tail and most recent knowledge (Mallen et al., 2022; Huang et al., 2022; Luu et al., 2022; Jang et al., 2022; Zheng et al., 2023). Retrieval-augmented language models (Izacard et al., 2022b; Borgeaud et al., 2022; Wang et al., 2023; Shi et al., 2023) have emerged as a powerful approach to address these issues by retrieving relevant knowledge from an external corpus. Among these, the encoder-decoder models, such as Atlas (Izacard et al., 2022b), stand out. They benefit from the strong representation ability of a bidirectional encoder, coupled with of the efficacy of a Fusion-in-Decoder architecture (Izacard & Grave, 2021), enabling the effective integration of multiple retrieved passages. Despite these advancements, in-context learning with these models remains underexplored.

---

[1]Code and model checkpoints will be made publicly available after the review process.

In this regard, we first conduct a comprehensive analysis of the state-of-the-art retrieval-augmented encoder-decoder language model, ATLAS, by experimenting with various prompting strategies. We find that ATLAS exhibits a certain in-context learning ability; however, due to a mismatch between pretraining and testing and a limited context length—issues that are common to existing encoder-decoder LMs trained with masked language modeling—its few-shot performance is not stable and providing more than, e.g., 8-shot, examples does not lead to further improvement.

Based on the analysis, we develop RAVEN[2] by first mitigating the mismatch between pretraining and testing of ATLAS through a combination of retrieval-augmented masked language modeling and prefix language modeling. Moreover, to enable the model to learn from more in-context examples, we propose *Fusion-in-Context Learning*, a novel approach that allows the model to utilize more in-context examples without modifying the model configuration or requiring additional training. Furthermore, we suggest using the retriever of the model to obtain relevant in-context examples to further enhance few-shot performance. Our empirical results demonstrate that RAVEN significantly outperforms ATLAS in both zero-shot and few-shot settings, even achieving comparable results to decoder-only large language models in some settings despite having 180 times fewer parameters.

The main contributions of this paper are summarized as follows:

- We present a comprehensive analysis of the in-context learning ability of the SOTA retrieval-augmented encoder-decoder language models and identify aspects for improvement.
- Based on the analytical groundwork, we develop RAVEN by combining retrieval-augmented masked and prefix language modeling.
- We further design *Fusion-in-Context Learning* and *In-Context Example Retrieval* to enhance the few-shot performance of retrieval-augmented encoder-decoder language models.
- We demonstrate the effectiveness of RAVEN and the proposed methods through extensive experiments, showcasing its superiority in various settings compared to strong baselines.

## 2 BACKGROUND AND RELATED WORK

**In-Context Learning.** In-context learning is one of the most significant features of LLMs (e.g., Dong et al., 2022). While there is growing interest in this area, most studies have focused on in-context learning with decoder-only LMs, e.g., GPT-3 (Brown et al., 2020). However, bidirectional LMs like BERT (Devlin et al., 2019) and T5 (Raffel et al., 2020) have been shown to achieve superior performance on various natural language understanding tasks, indicating that they may offer unique advantages for in-context learning as well. Patel et al. (2023); Chung et al. (2022) have initiated exploration into in-context learning with bidirectional LMs. While these studies have shown promising results, there is a considerable scope for further investigation. For instance, Patel et al. (2023) demonstrate that bidirectional models can outperform decoder-only LMs of a similar scale regarding in-context learning; however, there is still a significant performance gap compared to decoder-only models on a much larger scale.

**Retrieval-Augmented Language Models.** Retrieval-augmented language models are a class of language models designed to enhance their performance by incorporating external knowledge. These models typically employ an information retrieval mechanism to access relevant information from a large corpus, which is then integrated into the model's prediction process. Retrieval-augmented LMs can be based on both encoder-decoder (Izacard et al., 2022b; Lewis et al., 2020) and decoder-only (Khandelwal et al., 2020; Borgeaud et al., 2022; Shi et al., 2022) architectures. While there has been some research on in-context learning with retrieval-augmented decoder-only LMs, which can be straightforwardly implemented by concatenating retrieved passages with the query as the input of the LM (Mallen et al., 2022; Shi et al., 2023; Khattab et al., 2022), in-context learning with retrieval-augmented encoder-decoder LMs, such as ATLAS, remains unexplored to the best of our knowledge. Despite the fact that the encoder-decoder LMs can be more efficient at incorporating multiple (e.g., 40) retrieved passages (Izacard & Grave, 2021). In the following sections, we will start with an analysis of ATLAS and develop our model based on the analysis.

---

[2]RAVEN, a bird known for its intelligence and adaptability, has the letters "RA" in its name, which represents "**R**etrieval-**A**ugmented" in our context.

| Masked Language Modeling (Pretraining) | Prompting Strategy 1 | Prompting Strategy 2 |
|---|---|---|
| **Input to Encoder:**
Machine learning algorithms build a model based on sample data,<extra_id_0> as training data, in order to<extra_id_1> being explicitly programmed to do so. Machine learning algorithms are used in a wide variety of applications, such as in medicine, email filtering, speech recognition, agriculture, and computer vision,<extra_id_2> unfeasible to develop conventional algorithms to perform the<extra_id_3>
Passage: … machine learning models require a high quantity of reliable data in order for the models …

**Input to Decoder:**
*None*

**Output:**
<extra_id_0> known<extra_id_1> make predictions or decisions without<extra_id_2> where it is difficult or<extra_id_3> needed tasks. | **Input to Encoder:**
Question: What is the capital of the Provence-Alpes-Cote d'Azur region of France?
Answer: Marseilles
Question: The Greek word Xero (pronounced zero) in xerography and related terminology means what?
Answer: Dry
Question: In which country was the first permanent bungee jumping site situated?
Answer:<extra_id_0>
Passage: … first permanent commercial bungee site, the Kawarau Bridge Bungy at the Kawarau Gorge Suspension Bridge near Queenstown in the South Island of New Zealand …

**Input to Decoder:**
*None*

**Output:**
<extra_id_0> New Zealand | **Input to Encoder:**
Question: What is the capital of the Provence-Alpes-Cote d'Azur region of France?
Answer:<extra_id_0>
Question: The Greek word Xero (pronounced zero) in xerography and related terminology means what?
Answer:<extra_id_1>
Question: In which country was the first permanent bungee jumping site situated?
Answer:<extra_id_2>
Passage: … first permanent commercial bungee site, the Kawarau Bridge Bungy at the Kawarau Gorge Suspension Bridge near Queenstown in the South Island of New Zealand …

**Input to Decoder:**
<extra_id_0> Marseilles<extra_id_1> Dry

**Output:**
<extra_id_2> New Zealand |

Figure 1: Pretraining task of ATLAS and prompting strategies for in-context learning.

# 3 IN-CONTEXT LEARNING WITH ATLAS

ATLAS is the state-of-the-art retrieval-augmented encoder-decoder language model, which combines a general-purpose dense retriever and a sequence-to-sequence reader with the Fusion-in-Decoder architecture. The retriever, encoder and decoder are jointly trained during the pretraining process. In this process, the dense retriever, based on the Contriever model (Izacard et al., 2022a), is responsible for selecting relevant passages from an external knowledge source, e.g., Wikipedia, based on the given corrupted context. The retrieved passages are then processed along with the context by the encoder, which generates the corresponding output, i.e., the masked spans, at the decoder (Figure 1, left). ATLAS demonstrates exceptional few-shot performance on knowledge-intensive language tasks (Petroni et al., 2021), despite having a lower parameter count compared to other recent LLMs.

However, in Izacard et al. (2022b), the few-shot performance of ATLAS is achieved by finetuning the model with few-shot examples, which requires additional training and may limit its applications, such as dealing with dynamic and diverse real-time user queries like GPT-3/4 (Brown et al., 2020; OpenAI, 2023), where in-context learning plays a vital role. Nonetheless, the in-context learning ability of ATLAS has not been investigated in the original paper. Therefore, in this section, we aim to explore the in-context learning ability of ATLAS, using open-domain question answering (Chen et al., 2017) as a representative task.

## 3.1 PROMPTING STRATEGIES

To facilitate in-context learning, an effective prompting strategy is paramount. In contrast to decoder-only language models, where the input can only be fed to the decoder, encoder-decoder language models can take input in either the encoder or the decoder. In alignment with the pretraining objective of ATLAS, we identify two prompting strategies for in-context learning:

**Strategy 1.** The first strategy involves feeding all example question-answer pairs and the target question to the encoder, without any input to the decoder. The prompt is designed as:[3]

**Enc**: Question: $q_1$ Answer: $a_1 \ldots$ Question: $q_k$ Answer: $a_k$ Question: $q_0$ Answer:`<extra_id_0>` $d$

where $(q_1, a_1), \ldots, (q_k, a_k)$ represent example QA pairs, $q_0$ denotes the target question, `<extra_id_0>` is a sentinel token (Raffel et al., 2020), and $d$ is the relevant passage retrieved with $q_0$. An example in a 2-shot setting is illusated in Figure 1 (middle).

**Strategy 2.** As the decoder of ATLAS can also accept input, we can feed the answers of in-context examples to the decoder and only feed the questions to the encoder, using multiple sentinel tokens:

**Enc**: Question: $q_1$ Answer:`<extra_id_0>` $\ldots$ Question: $q_k$ Answer:`<extra_id_(k-1)>` Question: $q_0$ Answer:`<extra_id_k>` $d$

---

[3]Here we present a format designed for better demonstration. The actual prompt, which follows the template used in the pretraining of ATLAS, can be found in Appendix B.3.

Table 1: Results of Atlas 11B with prompting strategy 1 (S2) and strategy 2 (S2).

| | | Natural Questions | | | | TriviaQA | | | |
|---|---|---|---|---|---|---|---|---|---|
| | | 0-shot | 1-shot | 5-shot | 8-shot | 0-shot | 1-shot | 5-shot | 8-shot |
| Atlas | 11B S1 | 26.7 | 21.3 | 29.8 | **31.3** | 56.9 | 35.5 | 62.3 | **63.9** |
| Atlas | 11B S2 | | 21.4 | 16.3 | 9.8 | | 49.8 | 48.4 | 44.4 |

**Dec**: `<extra_id_0>` $a_1 \dots$ `<extra_id_(k-1)>` $a_k$

An example with this strategy is shown in Figure 1 (right). The model is expected to learn from in-context examples by examining both the input to the encoder and input to the decoder.

## 3.2 EXPERIMENTAL SETUP

We select two widely-used datasets in the domain of open-domain question answering: Natural Questions (NQ) (Kwiatkowski et al., 2019) and TriviaQA (TQA) (Joshi et al., 2017). To assess the performance, we follow the previous work (Izacard et al., 2022b) to employ the standard exact match (EM) metric. For the few-shot settings, we follow Brown et al. (2020) to evaluate each example in the test set by generating in-context examples through randomly sampling $k$ instances from the respective task's training set. Following Izacard et al. (2022b), we use an index composed of December 2018 Wikipedia dump for NQ and an index composed of December 2021 Wikipedia corpora for TriviaQA. We use the checkpoints released in the official repository[4], covering sizes of 11B (XXL), 3B (XL), and 770M (Large). We conduct experiments with various configurations detailed in the next section.

## 3.3 RESULTS & ANALYSIS

### 3.3.1 EFFECT OF PROMPTING STRATEGIES

We first study the effectiveness of the prompting strategies described in §3.1. Table 1 summarizes the results. We find that Atlas struggles to learn from in-context examples using strategy 2, as the few-shot performance is worse than the zero-shot performance. We hypothesize that this is because the model has difficulty learning the pattern of S2 with masked language modeling during its pretraining, since it is unlikely to obtain several consecutive question-answer pairs (or something similar) in the form of strategy 2 by randomly masking several spans in a sequence.

On the other hand, we observe that with strategy 1, the model does exhibit some in-context learning ability, where the 5-shot and 8-shot performance is significantly better than the zero-shot performance on both NQ and TriviaQA. Therefore, we choose to focus on strategy 1 for further study and disregard strategy 2 for the remainder of the paper.[5]

### 3.3.2 EFFECT OF NUMBER OF IN-CONTEXT EXAMPLES

The number of in-context examples is a crucial hyperparameter for in-context learning. Generally, we expect better performance from a model with more in-context examples, but there is an upper limit due to 1) the maximum context length setup, e.g., 512 tokens, during the pretraining process, and 2) the point at which the model has received sufficient examples and cannot gain additional information from more examples. The optimal number of in-context examples also varies between models. For instance, on TriviaQA, PaLM (Chowdhery et al., 2022) exhibits better 1-shot performance than settings with more examples, while this is not the case for GPT-3 (Brown et al., 2020).

Figure 2 illustrates the impact of varying the number of in-context examples across different Atlas model sizes. Interestingly, the 11B model demonstrates poor performance in low-shot settings, e.g., 1-shot, but improves significantly after 4-shot and 5-shot. Upon examining the generated responses, we find that the model tends to produce answers with more tokens in low-shot settings, while the ground truth typically consists of shorter answers with fewer than 5 tokens. By relaxing the criteria for a correct prediction to include instances where the ground-truth answer is a substring of the model output, we find that the 1-shot performance surpasses that of the 0-shot setting (38.3 vs 32.1 on NQ).

---

[4]`https://github.com/facebookresearch/atlas`
[5]We also study the effect of target question's position in Appendix C.1.

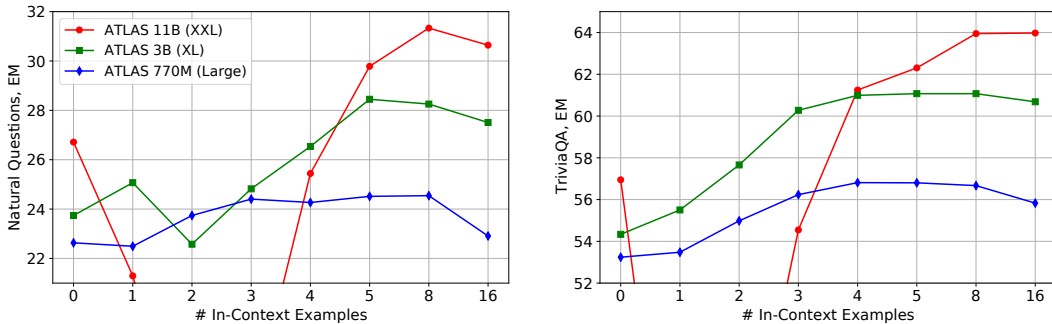

Figure 2: Results of ATLAS with different numbers of in-context examples.

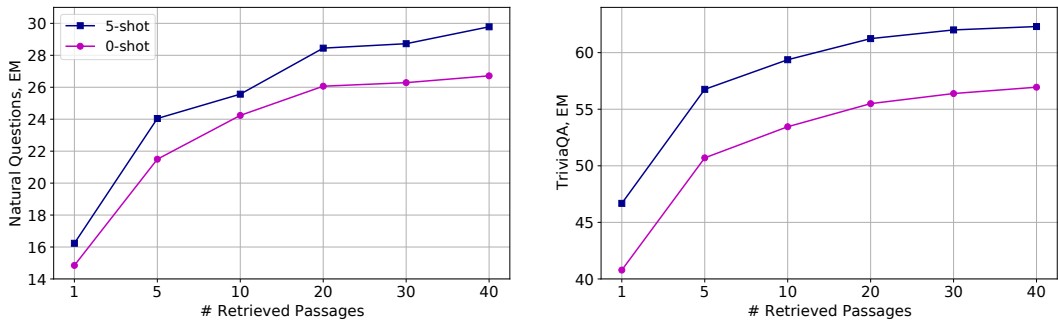

Figure 3: Results of ATLAS 11B with different numbers of retrieved passages.

All models perform well in the 5-shot and 8-shot settings, but their performance does not continue to improve with more in-context examples (e.g., 16-shot). We believe this plateau may be attributed to two factors: 1) the sequence length constraints during ATLAS pretraining, where the maximum input length to the encoder is set to 384 tokens, and the average input sequence length (excluding passages) is around 130 tokens; 2) the model's ability to learn adequately with 5 or 8 examples, making additional examples less beneficial.

### 3.3.3 EFFECT OF NUMBER OF RETRIEVED PASSAGES

Figure 3 illustrates the impact of the number of retrieved passages on model performance. We observe that for both 0-shot and 5-shot settings, the performance of the models increases significantly with the number of retrieved passages. This highlights the effectiveness of the Fusion-in-Decoder architecture (Izacard & Grave, 2021) for knowledge-intensive tasks like open-domain question answering, and underscores the importance of pretraining language models with retrieval augmentation.

Additionally, the 5-shot performance consistently outperforms the 0-shot setting. This observation further emphasizes the value of providing in-context examples to improve the performance of retrieval-augmented encoder-decoder language models.

### 3.3.4 SUMMARY

In summary, ATLAS exhibits a certain ability for in-context learning, which has been overlooked in previous studies. However, there are also some limitations such as unstable performance in low-shot settings, and the fact that providing more in-context examples does not consistently improve performance. Moreover, retrieving more relevant passages significantly enhances performance, demonstrating the significance of pretraining language models with retrieval for knowledge-intensive tasks. This also highlights the superiority of the encoder-decoder (Fusion-in-Decoder) architecture, which offers an advantage not available to decoder-only language models.

## 4 METHODOLOGY

In this section, we design methods to improve the models' zero-shot performance and in-context learning abilities based on the findings and analysis presented in §3.

## 4.1 RAVEN: COMBINING RETRIEVAL-AUGMENTED MASKED AND PREFIX LANGUAGE MODELING

As described in §3, ATLAS is pretrained with a masked language modeling objective, where the input is a corrupted text with several masked spans placed randomly within the sequence (refer to Figure 1 (left) for an example). However, in testing, based on our analysis in §3.3.1 and §C.1, it is most effective to place the target question after all the in-context examples, with a masked token (i.e., `<extra_id_0>`) following the question (Figure 1, middle)). Thus, there exists a mismatch between pretraining and testing of ATLAS.

To better align pretraining with testing, we propose to continually pretrain ATLAS with prefix language modeling (Liu et al., 2018; Raffel et al., 2020; Tay et al., 2023). Specifically, for each sequence, we mask 10% of the tokens on average at the end of the sequence with the `<extra_id_0>` token. Then, we use the retriever of ATLAS to retrieve relevant passages using the prefix and train the model to recover the suffix of this sequence with the prefix and the passages as

> **Prefix Language Modeling**
>
> **Input to Encoder:**
> Machine learning algorithms build a model based on sample data, known as training data, in order to make predictions or decisions without being explicitly programmed to do so. Machine learning algorithms are used in a wide variety of applications, such as in medicine, email filtering, speech recognition, agriculture, and computer vision, where it is difficult or`<extra_id_0>`
> Passage: … machine learning models require a high quantity of reliable data in order for the models …
>
> **Input to Decoder:**
> *None*
>
> **Output:**
> `<extra_id_0>` unfeasible to develop conventional algorithms to perform the needed tasks.

Figure 4: Prefix Language Modeling.

input. An example of input and output for prefix language modeling is shown in Figure 4. We can observe that the pretraining objective aligns more closely with the prompting strategy 1 in Figure 1. We refer to the model trained with additional prefix language modeling as RAVEN.

Starting from the ATLAS checkpoint, which is based on masked language modeling, the training of RAVEN can be considered a combination of retrieval-augmented masked and prefix language modeling. This methodology shares certain aspects with the mixture objective of UL2 (Tay et al., 2023). However, there are key differences: 1) UL2 blends various language modeling objectives throughout its training process, while RAVEN applies these two objectives in a sequential order; 2) Unlike RAVEN, UL2 is trained without retrieval. Consequently, RAVEN benefits from both the masked language modeling, which contributes to a better reader and retriever as evidenced in Izacard et al. (2022b), and prefix language modeling, which mitigates the gap between pretraining and testing. We verify the effectiveness of this design by exploring different training strategies in Appendix C.3.

## 4.2 FUSION-IN-CONTEXT LEARNING

In §3.3.2, we observe that ATLAS's performance does not further improve with more in-context examples after 8-shot. One major reason for this is the limited sequence length during the pretraining process, which makes it difficult for the model to handle long sequences during testing. Pretraining models with longer contexts would be a straightforward approach to address this issue, but it would significantly increase computation cost and GPU memory requirements. Additionally, the maximum input length is

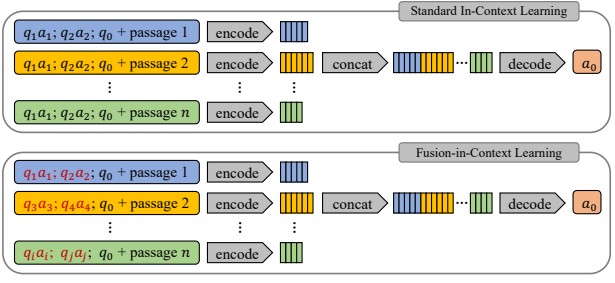

Figure 5: Standard In-Context Learning vs Fusion-in-Context Learning.

also constrained by the maximum sequence length of the retriever, i.e., Contriever, which is based on BERT (Devlin et al., 2019) and has a maximum length of 512 tokens.

As an alternative, we propose an approach to enable models to learn from more in-context examples without modifying the model configuration or requiring additional training. As described in §3, the reader of ATLAS (and RAVEN) is based on the Fusion-in-Decoder architecture (Izacard & Grave, 2021), where multiple passages are retrieved, and each passage, concatenated with the in-context examples and target question, is fed to the encoder separately (Figure 5, top). To allow the model to process more in-context examples without increasing the length of the input to the encoder, we can feed *different* in-context examples to the encoder with each passage (Figure 5, bottom). In this way,

Table 2: Results of ATLAS and RAVEN on NQ and TriviaQA.

| | | Natural Questions | | | | TriviaQA | | | |
|---|---|---|---|---|---|---|---|---|---|
| | | 0-shot | 1-shot | few-shot | FiCL | 0-shot | 1-shot | few-shot | FiCL |
| ATLAS | 3B | 23.7 | 25.1 | 28.4 (5) | 29.6 [64-5] | 54.3 | 55.5 | 61.1 (5) | 62.0 [64-5] |
| ATLAS | 11B | 26.7 | 21.3 | 31.3 (8) | 32.0 [64-8] | 56.9 | 35.5 | 63.9 (8) | 64.9 [64-8] |
| RAVEN | 3B | 29.3 | 31.7 | 31.4 (5) | 32.8 [40-1] | 62.4 | 63.2 | 62.6 (5) | 63.6 [40-1] |
| RAVEN | 11B | 29.6 | 31.4 | 32.7 (5) | **33.5** [64-5] | 65.7 | 66.2 | 66.7 (5) | **67.3** [64-5] |

the model can incorporate more in-context examples during its inference process. We refer to this strategy as *Fusion-in-Context Learning (FiCL)*.

In implementation, for a $k$-shot setting, such as a 64-shot setting, to effectively utilize the 64 examples, we randomly shuffle these examples and select $m$ (e.g., 5) examples in order as the input for the encoder each time. If all the examples have been used, we shuffle the 64 examples again. We denote the configuration of FiCL as $[k, m]$, which stands for [$k$-shot, $m$-fusion].

### 4.3 IN-CONTEXT EXAMPLE RETRIEVAL

In recent studies (Liu et al., 2022; Rubin et al., 2022; Su et al., 2023), it has been demonstrated that a well-chosen selection of in-context examples can enhance in-context learning. Building on this insight, we propose utilizing the retriever of RAVEN to retrieve in-context examples. Specifically, we use RAVEN's retriever to build an index during the preparation step, and then, during testing, when the model receives an input, it could efficiently retrieve in-context examples with its retriever.

By integrating RAVEN's retriever in this manner, we aim to: 1) automate in-context learning, which is particularly practical for model owners who have a database of examples. Without this, users would need to manually provide in-context examples; and 2) optimize the selection of in-context examples, thereby enhancing in-context learning and improving overall performance.

## 5 EXPERIMENTS

### 5.1 EXPERIMENTAL SETUP

**Datasets.** Following §3.2, we first evaluate on two widely-used open-domain question answering datasets: Natural Questions (Kwiatkowski et al., 2019) and TriviaQA (Joshi et al., 2017). Additionally, we conduct a case study on long-form question answering using the ELI5 dataset (Fan et al., 2019). Furthermore, we assess the models' language understanding ability using the Massively Multitask Language Understanding (MMLU) benchmark (Hendrycks et al., 2021). Detailed information regarding the MMLU evaluation is in Appendix B.4. Other evaluation settings are the same as §3.2.

**Baselines.** Since RAVEN is built upon ATLAS, we choose ATLAS as a primary baseline for comparison. We also compare our model with decoder-only large language models such as GPT-3 (Brown et al., 2020) and PaLM (Chowdhery et al., 2022) (in a closed-book setting). Additionally, for open-domain QA, we evaluate our approach against REPLUG (Shi et al., 2023) and RETRO (Borgeaud et al., 2022), as well as its improved version RETRO++ (Wang et al., 2023). These models are decoder-only language models augmented with retrieval. REPLUG is based on Codex (Chen et al., 2021) and Contriever (Izacard et al., 2022a), where the passages are retrieved by Contriever (using ensemble and additional adaptation) and fed directly to Codex. RETRO is a GPT model (Radford et al., 2019) augmented with a transformer encoder to encode the retrieved passages. RETRO++ is a variant of RETRO that feeds the most relevant retrieved passage into the GPT decoder while providing other passages to its encoder. For MMLU, we also include FLAN-T5 (Chung et al., 2022), an enhanced version of T5 that has been trained on a large mixture of tasks with instruction finetuning.[6]

### 5.2 OPEN-DOMAIN QUESTION ANSWERING

We choose open-domain QA as our primary evaluation task, as it effectively represents knowledge-intensive challenges and is widely employed in real-world applications.

---

[6]Implementation details are described in Appendix B.1.

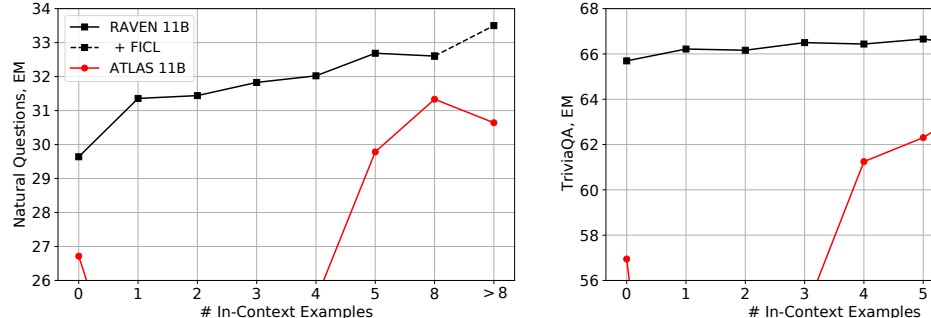

Figure 6: RAVEN vs ATLAS.

Table 3: Results on NQ and TriviaQA in comparison to the baselines.

| | | Natural Questions | | | TriviaQA | | |
|---|---|---|---|---|---|---|---|
| | | 0-shot | 1-shot | few-shot | 0-shot | 1-shot | few-shot |
| GPT-3 | 13B | 7.8 | 13.7 | 21.0 (64) | 41.8 | 51.3 | 57.5 (64) |
| GPT-3 | 175B | 14.6 | 23.0 | 29.9 (64) | 64.3 | 68.0 | 71.2 (64) |
| PaLM | 8B | 8.4 | 10.6 | 14.6 (5) | 39.5 | 48.5 | 47.2 (5) |
| PaLM | 62B | 18.1 | 23.1 | 27.6 (5) | 67.3 | 72.7 | 70.1 (5) |
| PaLM | 540B | 21.2 | 29.3 | 39.6 (64) | **76.9** | **81.4** | **81.4** (1)* |
| Codex | 175B | - | - | 40.6 (16) | - | - | 73.6 (16) |
| LLaMA | 7B | 16.8 | 18.7 | 26.1 (64) | 50.0 | 53.4 | 57.6 (64) |
| Codex + Contriever | 175B | - | - | 44.2 (16) | - | - | 76.0 (16) |
| Codex + REPLUG | 175B | - | - | 44.7 (16) | - | - | 76.8 (16) |
| Codex + REPLUG LSR | 175B | - | - | **45.5** (16) | - | - | 77.3 (16) |
| RETRO | 9.5B | 8.9 | - | - | 36.0 | - | - |
| RETRO++ | 9.5B | 25.8 | - | - | 48.3 | - | - |
| **RAVEN** | 3B | 29.3 | **31.7** | 31.4 (5) | 62.4 | 63.2 | 62.6 (5) |
| **RAVEN** + FiCL | 3B | | | 32.8 [40-1] | | | 63.6 [40-1] |
| **RAVEN** | 11B | **29.6** | 31.4 | 32.7 (5) | 65.7 | 66.2 | 66.7 (5) |
| **RAVEN** + FiCL | 11B | | | 33.5 [64-5] | | | 67.3 [64-5] |

\* For TriviaQA, PaLM's 1-shot performance surpasses other settings. We follow the original paper to report the 1-shot result.

For other models, we select the best $k$-shot ($k \in \{2, 3, 4, 5, 8, 16\}$) performance or report the number in the original paper.

**RAVEN VS ATLAS.** Table 2 and Figure 6 present the exact match (EM) scores for ATLAS and RAVEN on the NQ and TriviaQA datasets. As shown in Table 2, both the 3B and 11B RAVEN models significantly outperform ATLAS. For instance, on TriviaQA, RAVEN 11B achieves an improvement of 8.8%, 30.7%, and 2.8% in the 0-shot, 1-shot, and few-shot settings respectively, compared to ATLAS 11B. Furthermore, as illustrated in Figure 6, the performance of RAVEN increases steadily with the number of in-context examples, while the performance of ATLAS experiences a substantial decline in low-shot settings. These results demonstrate the effectiveness of RAVEN across various shot settings.

**Fusion-in-Context Learning.** We also report the results of models with Fusion-in-Context Learning (FiCL) in Table 2. For both ATLAS and RAVEN, FiCL contributes to approximately a 1% improvement, which is not attainable by standard in-context learning, where performance does not further improve (or even decreases) with more than 8 in-context examples. This demonstrates the superiority of FiCL for enabling models to learn from more examples.

**Comparison to SOTA.** In Table 3, we compare RAVEN to other baselines. On NQ, RAVEN's zero-shot and one-shot performance surpasses all the baselines, including PaLM, even though RAVEN 3B has 180 times fewer parameters than PaLM 540B. The zero-shot performance of RAVEN on TriviaQA is also on par with PaLM 62B. Furthermore, RAVEN's zero-shot performance significantly exceeds that of both RETRO and RETRO++, which are models of a similar scale.

In the few-shot setting, with FiCL, RAVEN achieves performance comparable to GPT-3 175B and PaLM 62B. However, there remains a gap between RAVEN and the larger PaLM 540B and Codex 175B models. Nevertheless, given the considerably smaller scale of RAVEN in comparison to PaLM

and Codex, its performance can be considered impressive. The performance of RAVEN may be further improved if it is built upon a larger model, in which case its few-shot performance is likely to surpass that of PaLM and Codex.

**In-Context Example Retrieval.** §4.3 suggests using RAVEN's retriever for in-context example retrieval. Results in Table 4 show that this approach improves RAVEN's few-shot results, especially on NQ where a ~10% improvement is observed. This indicates the significant positive impact of incorporating more relevant in-context examples.

**Additional Results.** We examine the **effect of the number of retrieved documents** in Appendix C.2, conduct an **ablation study of different training strategies** in Appendix C.3, and provide a **case study on long-form question answering** in Appendix C.4.

### 5.3 MMLU

Table 5 summarizes the results (accuracy) on Massive Multitask Language Understanding (MMLU). We find that the zero-shot performance of RAVEN is impressive, surpassing the few-shot performance of GPT-3 175B and being slightly worse than PaLM 62B, despite having a significantly smaller number of parameters. Furthermore, with the same number of parameters, the performance of RAVEN is far superior to T5. Additionally, even without instruction finetuning, RAVEN achieves performance comparable to FLAN-T5, a model finetuned on a large collection of tasks. We expect further improvement of RAVEN by applying instruction tuning as well and leave it for future study.

Interestingly, with standard in-context learning, the few-shot performance of RAVEN is worse than zero-shot, possibly due to the longer questions and answer options in MMLU causing context length issues in the 5-shot setting. Also, in the one-shot setting, since MMLU is a multiple-choice QA task, providing only one example might introduce bias in the model's prediction, favoring a specific option. However, with Fusion-in-Context Learning, the performance improves significantly, leading to better

Table 4: Performance improvement of RAVEN with In-Context Example Retrieval.

|  | NQ | | TQA | |
|---|---|---|---|---|
|  | 1-shot | 5-shot | 1-shot | 5-shot |
| 3B | +9.1 | +11.6 | +0.0 | +1.6 |
| 11B | +9.8 | +11.1 | -0.5 | +1.0 |

Table 5: Results on MMLU.

|  |  | 0-shot | 1-shot | 5-shot |
|---|---|---|---|---|
| GPT-3 | 13B | - | - | 26.0 |
| GPT-3 | 175B | - | - | 43.9 |
| PaLM | 8B | - | - | 25.3 |
| PaLM | 62B | - | - | 53.7 |
| PaLM | 540B | - | - | 69.3 |
| T5 | 3B | - | - | 25.7 |
| T5 | 11B | - | - | 25.9 |
| FLAN-T5 | 3B | - | - | 52.4 |
| FLAN-T5 | 11B | - | - | 55.1 |
| ATLAS | 3B | 43.7 | 36.9 | 38.5 |
| + FiCL | 3B |  |  | 42.6 [40-1] |
| ATLAS | 11B | 47.4 | 45.3 | 44.2 |
| + FiCL | 11B |  |  | 48.0 [40-1] |
| RAVEN | 3B | 45.7 | 40.0 | 40.4 |
| + FiCL | 3B |  |  | 44.5 [64-5] |
| RAVEN | 11B | 48.9 | 49.2 | 48.7 |
| + FiCL | 11B |  |  | 50.5 [40-1] |

few-shot performance for the 11B model compared to its zero-shot performance, further demonstrating the effectiveness of FiCL.

## 6 CONCLUSION

In this study, we have delved into the in-context learning ability of retrieval-augmented encoder-decoder language models. We commenced with a comprehensive analysis of the state-of-the-art ATLAS model and subsequently developed our model based on the analysis. Our extensive experimental results demonstrated that our model significantly outperforms ATLAS and achieves results on par with some of the most advanced language models, even with substantially fewer parameters. These findings highlight the potential of retrieval-augmented encoder-decoder language models in the realm of in-context learning.

Although we started with ATLAS, the insights and proposed methods are transferrable and can potentially enhance other models, such as domain-specialized or more powerful ones. The training strategy of RAVEN and the idea of FiCL are simple yet effective, and would not have been conceived without our analytical groundwork. Future work focusing on scaling up the model, applying these methods, and further studying its in-context learning ability is encouraged.

REPRODUCIBILITY STATEMENT

We have detailed the hyperparameter setup and configurations in Appendix B.1. The checkpoints for the ATLAS and T5 models used in our experiments are publicly available at `https://github.com/facebookresearch/atlas` and `https://huggingface.co/google/t5-xl-lm-adapt`, respectively. The exact input prompts for both pretraining and testing are included in Appendix B. The full code and model checkpoints will be made publicly available after the review process.

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

## A    LIMITATIONS AND BROADER IMPACT

One primary limitation of our work arises from the constrained context length inherent to the base models (e.g., T5 or ATLAS) we employed. This restriction poses challenges to the scalability of in-context learning, especially as the number of in-context examples increases. While our Fusion-in-Context Learning (FiCL) strategy does offer a mitigative approach to this constraint, an alternative and possibly more optimal solution might involve extending the context length. This would be particularly beneficial for tasks requiring extensive inputs.

Furthermore, when compared to some of the prevailing decoder-only language models, particularly those exceeding 100B parameters, the models deployed in our research might appear relatively diminutive in scale. While our starting point was ATLAS, the insights and methods we have uncovered are transferrable and can potentially enhance other models, including domain-specialized or more powerful ones. Our endeavor partially seeks to catalyze further investigations into more powerful encoder-decoder models. Drawing on the benefits of scaling up and combining this with our proposed approaches, we believe that there is potential to develop even more powerful retrieval-augmented language models in the future.

## B    ADDITIONAL EXPERIMENTAL DETAILS

### B.1    TRAINING DETAILS

We train two versions of RAVEN: 3B and 11B. We initialize both the retriever and the reader of the models with the weights of ATLAS (3B and 11B). To isolate the effect of retrieval, we do not update the retriever during the training process. We pretrain the reader using the December 2021 Wikipedia corpora preprocessed by Izacard et al. (2022b), where the index is also constructed using the same corpora. In accordance with Izacard et al. (2022b), we retrieve 20 passages for each masked sequence (excluding passages identical to the original sequence). Both the 3B and 11B models are trained for 5,000 steps, using AdamW optimizer (Loshchilov & Hutter, 2019) with a batch size of 64. We employ a learning rate of $4 \times 10^{-5}$ for the 3B model and $1 \times 10^{-5}$ for the 11B model, with linear decay and 100 warmup steps. All the models are trained on NVIDIA A100 GPUs (80 GB). For the 3B model, we utilize 8 GPUs, whereas for the 11B model, we employ 32 GPUs. The prompt used for prefix language modeling is detailed in Appendix B.2. During testing, we default to retrieving 40 documents for all tasks. The prompts used can be found in Appendix B.3 and Appendix B.4.

### B.2    PREFIX LANGUAGE MODELING

In alignment with the pretraining of ATLAS, we design the prompt for prefix language modeling as

```
{prefix}<extra_id_0> title: {title} context: {text}
```

where {prefix} represents the prefix of an input sequence. The {title} and {text} elements are retrieved by the model's retriever using the prefix as a query. Here, {text} signifies the retrieved passage, while {title} denotes the corresponding article and section title of the passage. The model is trained to generate

```
<extra_id_0>{suffix}
```

where {suffix} is the suffix (masked by `<extra_id_0>`) of the input sequence.

### B.3    OPEN-DOMAIN QUESTION ANSWERING

In accordance with pretraining, we use the following prompt for open-domain question answering:

```
Question: {question} Answer:<extra_id_0> title: {title} context:
{text}
```

For example,

```
Question: In which country was the first permanent bungee jumping
site situated? Answer:<extra_id_0> title: Bungee jumping: Modern
```

```
sport context:  first permanent commercial bungee site, the
Kawarau Bridge Bungy at the Kawarau Gorge Suspension Bridge
near Queenstown in the South Island of New Zealand.  Hackett
remains one of the largest commercial operators, with concerns
in several countries.  Several million successful jumps have
taken place since 1980.  This safety record is attributable to
bungee operators rigorously conforming to standards and guidelines
governing jumps, such as double checking calculations and fittings
for every jump.  As with any sport, injuries can still occur (see
below), and there have been fatalities.  A relatively common
mistake in fatality cases is to use a cord that
```

### B.4 MASSIVE MULTITASK LANGUAGE UNDERSTANDING

MMLU comprises 57 multiple-choice question answering datasets that span various domains, including elementary mathematics, US history, computer science, and more. For the evaluation on MMLU, we report the accuracy and use an index composed of December 2021 Wikipedia corpora. We follow Izacard et al. (2022b) to apply the "de-biased" inference. Specifically, during inference, we execute four forward passes, each corresponding to a cyclic permutation of the answer letter-option assignment within the question. For instance, the answer option designated to letter 'A' is shifted to 'B', 'B' to 'C', 'C' to 'D', and 'D' to 'A'. The final prediction is obtained by summing up the probabilities from these four forward passes.

We design the prompt in the following format:

```
Question:  {question} Options:  {candidate answers}
Answer:<extra_id_0> title:  {title} context:  {text}
```

For example,

```
Question:  Over time, non-volcanic mountains can form due to the
interaction of plate boundaries.  Which interaction is most likely
associated with the formation of non-volcanic mountains?  Options:
(A) continental plates colliding with continental plates (B)
continental plates separating from continental plates (C) oceanic
plates colliding with oceanic plates (D) oceanic plates separating
from oceanic plates Answer:<extra_id_0> title:  ...  context:  ...
```

Given that many questions in the MMLU benchmark are quite lengthy, concatenating in-context examples (questions and candidate answers) with the target question in a few-shot setting is likely to exceed the maximum input length. To mitigate this, we only sample examples with question lengths of fewer than 50 tokens to use as in-context examples.

## C ADDITIONAL RESULTS

### C.1 EFFECT OF POSITION (ATLAS)

As ATLAS is an encoder-decoder language model with a bidirectional encoder, it can also examine in-context examples that follow the target question to fill in the masked token. This means that we may position the target question at the beginning or middle of a sequence, for example:

Question: $q_0$ Answer:`<extra_id_0>` Question: $q_1$ Answer: $a_1$ ... Question: $q_k$ Answer: $a_k$ $d$

Question: $q_1$ Answer: $a_1$ ... Question: $q_0$ Answer:`<extra_id_0>` ... Question: $q_k$ Answer: $a_k$ $d$

Table 6: Results of ATLAS 11B (5-shot) with different target question positions.

|  | NQ | TQA |
|---|---|---|
| *first* | 0.7 | 9.2 |
| *random* | 6.5 | 19.5 |
| *last* | **29.8** | **62.3** |

Table 6 summarizes the results. We denote the target question's position as "*first*" for the beginning of the sequence, "*random*" for a random position, and "*last*" for the original setting (S1). Interestingly, placing the target question anywhere other than the last position results in a significant performance drop. Upon examining the generated answers, we observe that when the target question is placed

at the beginning or in the middle, the model tends to repeat the answer or generate additional text. For example, for the prompt "Question: What number in Bingo is sometimes referred to as Heinz varieties? Answer:<extra_id_0> Question: ...". The generated text is "57 'Heinz varieties' is a term used in Bingo to describe". This indicates that the model does not fully understand and follow the style of in-context examples. Therefore, by default, we position the target question after all the in-context examples.

## C.2 EFFECT OF NUMBER OF RETRIEVED PASSAGES (RAVEN)

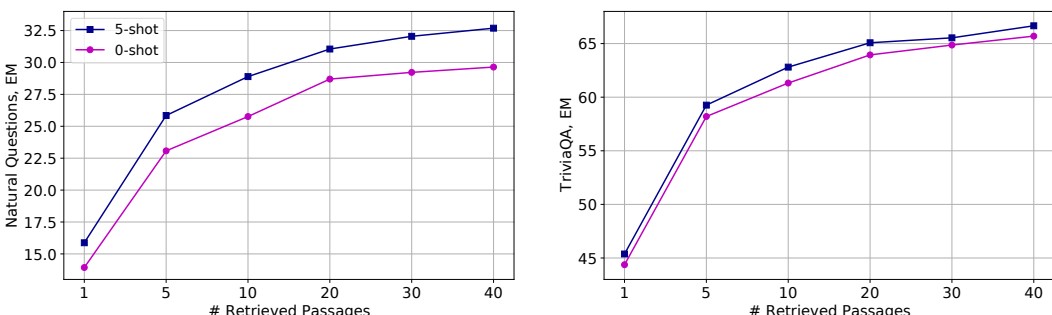

Figure 7: Results of RAVEN 11B with different numbers of retrieved passages.

Figure 7 illustrates the effect of the number of retrieved passages on the performance of RAVEN 11B. The model performance improves with an increased number of retrieved passages in both 0-shot and 5-shot settings, with the 5-shot performance consistently outperforming 0-shot. These observations align with the ones presented in §3.3.3.

## C.3 ABLATION STUDY

Table 7: Results of ATLAS and RAVEN trained with different strategies.

|  |  | **Natural Questions** | | | **TriviaQA** | | |
|---|---|---|---|---|---|---|---|
|  |  | 0-shot | 1-shot | 5-shot | 0-shot | 1-shot | 5-shot |
| ATLAS | 3B (Mask) | 23.7 | 25.1 | 28.4 | 54.3 | 55.5 | 61.1 |
| ATLAS | 3B (Mask, 5k more steps) | 22.9 | 22.5 | 28.1 | 50.8 | 50.1 | 61.1 |
| RAVEN⁻ | 3B (Prefix) | 24.8 | 29.1 | 30.1 | 55.4 | 61.4 | 62.3 |
| RAVEN⁻ | 3B (Mix) | 25.1 | 28.4 | 30.9 | 56.1 | 61.4 | 62.2 |
| RAVEN | 3B | 29.3 | 31.7 | 31.4 | 62.4 | 63.2 | 62.6 |

We conduct an ablation study by training ATLAS and RAVEN with different pretraining strategies. First, to isolate the effect of more training steps of RAVEN, we also train ATLAS for 5,000 more steps using the masked language modeling objective. Results in Table 7 (row 2) show that the performance does not improve, indicating that the performance improvement of RAVEN compared to ATLAS is not simply due to training for more steps.

Second, to verify the effectiveness of RAVEN's training strategy (i.e., first masked language modeling, and then prefix language modeling), we train two variants of RAVEN, starting from the T5-lm-adapt checkpoint[7], which is the checkpoint that ATLAS starts from. For the first variant, we use the same prefix language modeling objective of RAVEN. For the second variant, we train the model with a mixture of masked and prefix language modeling. Specifically, we construct corrupted texts by both masking 15% spans in the sequence (same as ATLAS) and replacing the suffix with a special mask token <extra_id_99> (used in testing). We train the model for 10,000 steps and update the retriever and refresh the index during training with the optimal strategy described in Izacard et al. (2022b). Table 7 (RAVEN⁻ in row 3 and 4) summarizes the results. We find that the performance of these two variants is superior to ATLAS, but inferior to RAVEN when trained using the strategy

---
[7]https://huggingface.co/google/t5-xl-lm-adapt

described in §4.1. An explanation for this is that, by training with masked language modeling first, the model can achieve better language understanding ability and is equipped with a more effective retriever (as emperically verified in Izacard et al. (2022b)). Subsequently, by training with prefix language modeling, the mismatch between pretraining and testing is mitigated, resulting in improved zero-shot and few-shot performance.

### C.4 LONG-FORM QUESTION ANSWERING

Table 8 presents some example outputs of ATLAS and RAVEN 11B on long-form question answering. The questions are sampled from the ELI5 dataset (Fan et al., 2019). An examination of these results reveals that ATLAS typically generates concise answers, while the output from RAVEN generally encompasses more information. This is a predictable outcome given that ATLAS is pretrained solely with masked language modeling, where each masked span usually contains only a handful of tokens. Besides, while RAVEN's answers are not always entirely accurate, they generally exhibit higher quality compared to ATLAS. Furthermore, the use of Fusion-in-Context Learning in RAVEN appears to contribute to a more coherent and informative generation.

Table 8: Example outputs of ATLAS and RAVEN 11B on long-form question answering.

| Question | How does a painting like "When Will You Marry" sell for $300 million dollars when it doesn't look like anything spectacular |
|---|---|
| Human | When it comes to a lot of art, context is more important than the actual material. Rarity, historical context etc. make a thing more valuable than its actual aesthetic properties. For example, some shitty buggy SNES or N64 games will sell for huge amounts if they are well preserved and in the original packaging etc, even when any high school student nowadays could write a better game in a week. |
| ATLAS 1-shot | It's a masterpiece. |
| RAVEN 1-shot | It's not the painting that sells for $300 million dollars, it's the story behind the painting. |
| RAVEN FiCL [40-1] | It's not the painting that sells for $300 million dollars, it's the reputation of the artist. |

| Question | When the society goes fully automatized, what will be the "normal" people needed for? |
|---|---|
| Human | Call centers... I am being totally serious. If your Sony TV is being completely fabricated by machines, shipped by machines, sold on amazon and delivered by machines, one of the few value adds companies will be able to deliver is GOOD customer service. So calling that 1-800 number and being instantly connected to a knowledgeable, friendly, human being who is going to patiently help you fix whatever issue you have (FYI... you forgot to plug it in). |
| ATLAS 1-shot | '> ELI5: What will the society go fully automatized? > |
| RAVEN 1-shot | The ones who will program the robots. |
| RAVEN FiCL [40-1] | The "normal" people will be needed to program the robots. |

| Question | Why is a 2 liter of Coca-cola $1, a 12 oz bottle $1.49 and the same size bottle of water $1.99? |
|---|---|
| Human | Because people will pay. Plain and simple. Capitalism at its best. |
| ATLAS 1-shot | The price of the product is based on the cost of the packaging. |
| RAVEN 1-shot | The cost of packaging is proportional to the size of the container. |
| RAVEN FiCL [40-1] | The cost of packaging is the largest portion of the cost of a product. The larger the container, the less packaging is required. |

| Question | If Matter Cannot Be Created or Destroyed... How do trees form? |
|---|---|
| Human | Say you have a box of Legos with 500 pieces. With those 500 pieces, you build a bulldozer. Then one day, you decide to rearrange those pieces from a bulldozer to a monster truck. It's still the same 500 pieces, just in a different layout. This is how trees are formed. The matter is just rearranged from other objects. Soil, nutrients, water, and even air. This is called "Conservation of Mass." EDIT: added air. Thanks. :) |
| ATLAS 1-shot | Trees are made of carbon and oxygen. |
| RAVEN 1-shot | Matter can be created or destroyed by combining two or more different types of matter. |
| RAVEN FiCL [40-1] | The tree is a living organism. It is made up of matter. The tree is made up of carbon, hydrogen, oxygen, and water. |

| Question | If the flu shot contains a non-active version of the flu, then why are "flu like symptoms" common for the days following? |
|---|---|
| Human | the non-active proteins trigger an immune system response which can involve some inflammatory system responses. once the immune system is activated for that infection your body will react more readily if the actual infection gets into you. |
| ATLAS 1-shot | The flu shot contains live virus particles. |
| RAVEN 1-shot | The flu shot contains a live attenuated virus. This means that it is not able to infect the body, but it can still cause some side effects. |
| RAVEN FiCL [40-1] | The flu shot contains a weakened version of the virus. This weakened virus is able to trigger an immune response in the body. This immune response can cause flu like symptoms. |

