# OpenReview forum: "In-Context Learning with Retrieval Augmented Encoder-Decoder Language Models"
_ICLR.cc/2024/Conference — ICLR 2024 Conference Withdrawn Submission_

### Official Review · Reviewer_2TC1 · 2023-10-28

**Soundness:** 3 good
**Presentation:** 3 good
**Contribution:** 2 fair
**Rating:** 5
**Confidence:** 4

**Summary:**

This paper studies the retrieval-augmented language model of encoder-decoder architecture. Specifically, this paper bridges the gap between pre-training and inference of the sota retrieval-augmented encoder decoder language model ATLAS by combining retrieval-augmented masked and prefix language modeling. Then, this paper designs Fusion-in-Context Learning and In-Context Example Retrieval to enhance the few-shot performance of retrieval-augmented encoder-decoder language models. Experimental results show the proposed method is effective.

**Strengths:**

1. Propose an interesting question “in-context learning in retrieval augmented language model”. Decoder-only language models are good at in-context learning while hard to use retrieved information. Encoder-decoder language models are good at retrieval augmented generation due to their separated encoder while unstable for in-context learning. This paper aims to introduce in-context learning ability into encoder-decoder language models.
2. This paper is well-written and easy to follow. The proposed method is well-motivated and very easy to implement.
3. Detailed experimental results on diverse datasets and the additional analysis can answer the main concern of motivation.

**Weaknesses:**

1. Although I admit that exploring the in-context learning (ICL) in retrieval-augmented language model is an interesting research question, I think the specific research points of this paper are still limited. This paper only study how to make the language model perform better on question-answering task under the retrieval-augmented paradigm. This is inconsistent with our accepted definition and expectation of ICL. ICL enables the language model to learn and perform various tasks using a few input examples. Just focusing on knowledge-intensive tasks such as question-answering cannot be called ICL. It is more like few-shot learning for question-answering.
2. The technical contribution of this paper is limited.
 - In order to bridge the gap between pre-training and inference in ATLAS, this paper introduces prefix language modeling. It just puts the special token at the end of the texts. Prefix language modeling is a well-known method in the pre-training of decoder-only language models such as GPT.
 - In order to input more examples to the retrieval-augmented language model, this paper just feeds different examples to FID encoder with each passage.
The above two points are the core technical contribution of this paper which is limited.
3. This paper emphasizes the advantages of the encoder-decoder architecture model in retrieval-augmented compared to the decoder-only model, but its core technical contribution is to change the masked language modeling of the encoder model to the prefix language modeling of the decoder-only model. This point is very contradictory.
4. Some decoder-only pre-trained language models with considerable size such as LLama-7B should be used as baselines. Many papers show LLama-7B can achieve better performance in Table 3. I hope the author can carefully analyze the advantages of the encoder-decoder architecture model compared to the decoder-only model in terms of retrieval-augment.

**Questions:**

Please see the weakness section.

Additional questions:
1. In Figure 6, the x-axis label '>8' should be a precise number instead of a span.
2. The efficiency of the Raven should be taken into consideration. Especially comparing it to the baselines.
3. Please show the differences between Raven and "FiD-ICL: A Fusion-in-Decoder Approach for Efficient In-Context Learning".

---

> ### Author Response · Authors · 2023-11-18
> **Response to Reviewer 2TC1 [1/2]**
>
> We thank the reviewer for highlighting the motivation and soundness of our study, as well as the writing of our paper. We greatly appreciate your thorough review and constructive feedback. We are pleased to address the points you raised.
>
>
> >  **Scope of In-Context Learning**:
>
> Thank you for your comment. We test on both open-domain question answering and MMLU, as well as providing case studies on long-form question answering. These are common tasks used for testing the performance of retrieval-augmented LMs in other studies, such as Shi et al., (2023).
>
> As our training objectives (masked and prefix language modeling) are not directly optimized for question answering, they should generalize to various tasks.
>
> > **Technical Contribution**:
>
> Our work offers contributions from both analytical and technological perspectives:
> - From an analytical standpoint, we provide a thorough analysis of the in-context learning capability of state-of-the-art retrieval-augmented encoder-decoder language models. Our insights also highlight potential avenues for improvement. This analysis serves as the foundation for our proposed RAVEN training strategy and Fusion-in-Context prompting approach.
> - From a technological perspective, we introduce a confluence of retrieval-augmented masked and prefix language modeling, coupled with our Fusion-in-Context Prompting and In-Context Example Retrieval strategies. These techniques not only enhance the base model's capabilities (by mitigating the mismatch between pretraining and testing) but also maximize the potential of in-context learning for these models (either by scaling the number of in-context examples with FiCL or by retrieving better in-context examples).
>
> It's crucial to emphasize that our methodologies weren't spontaneously conceived; they stem from an in-depth understanding of the limitations of current models and a careful consideration of how to overcome these issues as discussed in Sections 3 and 4. While the technique's design might appear simple, its conception was far from obvious without our analytical groundwork. Moreover, simplicity (when effective) is a commendable attribute of a technique.
>
> > **Contradictory Points on Encoder-Decoder Architecture**:
>
> Here, we would like to correct a misconception of the reviewer: prefix language modeling is not exclusive to decoder-only language models. The distinction between encoder-decoder LMs and decoder-only LMs lies in their architecture, not in their pretraining objectives (the reviewer may refer to UL2 (Tay et al., 2023) for more details). However, a significant feature that decoder-only LMs lack is the bidirectional architecture. As noted in the GPT-3 paper: “GPT-3 has several structural and algorithmic limitations ... as a result, our experiments do not include any bidirectional architectures or other training objectives such as denoising ... our design decision comes at the cost of potentially worse performance on tasks which empirically benefit from bidirectionality …” Our method capitalizes on the strengths of both masked and prefix language modeling with the incorporation of a bidirectional architecture.
>
> >**Comparison with Decoder-Only Models**:
>
> Thank you for your comments. We have compared our model with RePlug (based on Codex) and Retro, which are decoder-only language models augmented with retrieval. Retro, having a parameter count similar to our models, performs significantly worse than ours. We have included the results of LLaMA 7B in Table 3. We did not find its performance superior and would be interested in knowing the sources of the papers that suggest otherwise.
>
> Here, we also want to emphasize that this comparison may not be very meaningful, as the models are trained on different data. It is unrealistic to expect a model developed with an academic budget to match the performance of larger-scale models (where "larger-scale" refers to both the number of parameters and the extent of training data), although our results are impressive considering the scale and budget.
>
> Furthermore, our aim is not to build the SOTA model, as this is impossible considering the budget, and no model stronger than GPT-4 truly exists yet. Our main goal is to demonstrate the possibility of in-context learning with retrieval-augmented encoder-decoder language models as a research practice. We provide comprehensive analysis and design techniques that enable these models to perform better in in-context learning.

---

> ### Author Response · Authors · 2023-11-18
> **Response to Reviewer 2TC1 [2/2]**
>
> > **Additional Questions**:
>
> Thank you for your questions. We respond to them one by one below:
>
> 1. Here, we report the best observed few-shot performance with more than 8 shots.
> 2. For decoder-only LMs, the computational cost typically increases quadratically with the input length, as well as with the number of retrieval passages. In contrast, for encoder-decoder LMs with a fusion-in-decoder architecture, the computation cost grows linearly with the number of retrieved passages, as they only perform self attention over one passage at a time (Izacard et al., 2021). Therefore, from the perspective of computational overhead increasing with the number of retrieved passages, retrieval-augmented encoder-decoder language models are superior to decoder-only language models.
> 3. This work was released after we completed our draft. The differences include: 1) As depicted in their Figure 1 (middle), they encode each example/input separately. In contrast, RAVEN encodes multiple examples along with the target input and each retrieved passage. 2) Their study does not focus on retrieval-augmentation. Although the title appears relevant to our proposed FiCL method, the content of the work significantly differs from our work.
>
> Thank you for your detailed comments. We hope our response addresses your questions. Please let us know your thoughts, and we are more than happy to answer any further questions.

---

> > ### Comment · Reviewer_2TC1 · 2023-11-21
> >
> > Thanks a lot for the detailed response, it addresses many concerns in my review.
> >
> > However, I still have some questions that confuse me a lot and they result in keeping the current slight negative score.
> >
> > 1. As the experimental results in MMLU,
> > - Firstly, we can find that k-shot learning and FiCL do not work for this dataset, for example, they are worse than the zero-shot performance for Atlas 3B and Raven 3B models and very few improvements in 11B models.
> > - Secondly, the statement "since MMLU is a multiple-choice QA task, providing only one example might introduce bias in the model’s prediction, favoring a specific option" is not convincing, because the question-answering problem also can introduce bias that the model prefers some types of answer. If this is the case, why not use 2-shot settings?
> > - Thirdly, the zero-shot and one-shot of GPT-3 and T5 are missing without any reasonable explanation.
> >
> > 2. I find that the configurations in Table 2 are very different for different models, for example, the few-shot for Atlas 11B is set to 8 and others are 5, Raven 3B FiCL setting is different from others.
> >
> > 3. I think LLama 7B is not too large to use as a baseline, because authors also use 11B models as their backbone models.

---

> ### Author Response · Authors · 2023-11-21
> **Thanks for acknowledging our response**
>
> Thank you for acknowledging our response! We are glad that our response addressed many of your concerns, and we answer your follow-up questions below.
>
> > **1. MMLU**
>
> The experiments in MMLU demonstrate the effectiveness of FiCL by comparing the few-shot results with and without FiCL. This aspect is significant, as we can observe a ~4% improvement.
>
> It’s true that the selection of in-context examples also affects question answering (Table 4 of our paper demonstrates this a bit). However, the bias is more pronounced and negative in multiple-choice problems, which is why FiCL here is very effective, as it mitigates bias without requiring longer context. The 2-shot setting would be interesting. However, as the discussion period is ending soon, we hope the reviewer can understand that we may not have enough time to run the experiments. The reason we report 5-shot rather than 2-shot results is simply because previous works use the 5-shot setting for MMLU and do not include results for 2-shot.
>
> Since previous works report only 5-shot results, we follow them to report the 5-shot results. For GPT-3 and T5, the 5-shot performance is not strong, so reporting the additional 0-shot and 1-shot performance is not very meaningful. Moreover, as mentioned in our previous response, the comparison is not very meaningful, and our primary goal is not to build the SOTA model, where scaling and data engineering is usually the best approach. **We regard the results of these models more as references to gauge our model's position. We treat this work as a pioneer effort to demonstrate the *possibility* and *potential*, and we encourage future work to scale up the model in our conclusion section.**
>
> > **2. Configurations in Table 2**
>
> For results without FiCL, we explored shots from 1, 2, 3, 4, 5, 8, 16, 32, 64, and reported the best observed results; for results with FiCL, based on few-shot results, we explored three significant combinations: [64-1] (or [40-1]), [64-5], [64-8].
>
> This is a common practice for LLM papers reporting few-shot performance, as observed in Table 3 where the best results of other models are based on different shots (e.g., For PaLM, the reported results include 1, 5, 64-shot). And in fact, RAVEN's performance across various shots is stable. The reviewer may refer to Figure 6 again, which consistently shows RAVEN's few-shot performance significantly outperforming that of ATLAS across different shots.
>
> > **3. LLama 7B**
>
> Yes, we added the results of LLama 7B to Table 3. We would also be happy to report the results of LLama 7B with retrieval augmentation, but we did not find it in existing literature. Instead, we report results of RePlug and Retro, which are retrieval-augmented models based on decoder-only language models.
>
> Here, the scale is not just about model parameters but also about the number of training data and the quality of training data. The comparison between different base models is not very meaningful. A better base model can no doubt achieve better performance. Whether it outperforms or underperforms, it does not diminish the contribution of the paper. The results serve more as a reference, related to engineering effort rather than research effort, as scaling and data engineering is usually the best practice if the goal is to achieve better performance. And it’s somewhat meaningless to care about this kind of comparison, as GPT-4 is always a stronger baseline, lol.
>
> **We hope the reviewer can regard this work more as a research practice that shows the *possibility* and *potential*, focusing on the analysis, design, and comparison between RAVEN and ATLAS.**
>
> We hope our response addresses your questions, and we are more than happy to answer any further questions.

---

> > ### Comment · Reviewer_2TC1 · 2023-11-22
> >
> > The issue arises when my focus narrows down to "this work more as a research practice that shows the possibility and potential, focusing on the analysis, design, and comparison between RAVEN and ATLAS". Consequently, it seems that this study may be perceived as too confined, echoing the concerns raised by Reviewer y5kL. Given that ATLAS represents merely one encoder-decoder model, the proposed methodology's exclusive testing on this specific model could potentially limit its scope for future applications.

---

> > > ### Author Response · Authors · 2023-11-22
> > > **Thank you for your reply**
> > >
> > > Thank you for your reply! Both ATLAS and RAVEN are essentially based on the T5 model, which could be substituted with other variants like mT5 (Xue et al., 2021) and UL2 (Tay et al., 2023). Therefore, the proposed methods can be applied across this family of models. In Appendix C.3, we explore different training strategies starting from T5, demonstrating the effectiveness of our design.
> > >
> > > I believe the main argument here concerns the family of models: decoder-only LMs are more popular than encoder-decoder LMs today. This leads to the perception that a method designed for encoder-decoder LMs is seen as confined, while one for decoder-only LMs is not. However, from a research standpoint, exploring something unique and demonstrating potential is crucial. We also hope our research will encourage the development of more powerful encoder-decoder language models—a domain we believe has great potential but is relatively less studied today. The performance of RAVEN could also be further enhanced when based on a more advanced base model.
> > >
> > > We would like to share these thoughts and are happy to continue the discussion. We also thank the reviewer for the insightful feedback and respect the viewpoint.

---

### Official Review · Reviewer_y5kL · 2023-10-28

**Soundness:** 2 fair
**Presentation:** 3 good
**Contribution:** 2 fair
**Rating:** 5
**Confidence:** 2

**Summary:**

The paper is built on the work of ATLAS.
The paper explores the drawbacks of ATLAS in the aspect of in-context learning (ICL) and identifies two issues of ICL with ATLAS: (1) mismatch between pretraining and testing and (2) restricted context length.
To tackle the first issue, the paper proposes to continually pretrain ATLAS masking 10% of the tokens at the later part of the sequence.
To tackle the second issue, the paper proposes to blend limited in-context examples into Fusion-in-Decoder architecture to avoid increasing the length of the input to the decoder.
The paper finally shows the proposed method outperforms ATLAS and has comparable performance to decoder-only LLMs.

**Strengths:**

(1) The explored topic, combining in-context learning (ICL) and retrieval augmented generation (RAG) is very interesting.

(2) The paper is well-written, and the proposed idea is simple and easy to follow.

(3) The paper identifies two drawbacks of ATLAS on ICL, including (1) the mismatch between pretraining and testing and (2) the restricted context length, which makes sense to me.

(4) Compared with ATLAS, the performance is significantly improved.

**Weaknesses:**

(1) Though the proposed method has a significantly better few-shot performance than ATLAS, the work seems to be incremental compared with ATLAS in the aspect of the modeling. The first method is to continuously pretrain ATLAS by masking the later part of the sequence. The second method is to blend different limited in-context examples into different passages in the Fusion-in-Decoder architecture. The whole architecture is exactly the same as ATLAS, which makes me feel the proposed method is more like the tricks for ATLAS for better ICL performance, including a fine-tuning method and a prompt engineering method.

(2) Though the proposed method outperforms ATLAS, it still cannot match decoder-only LLM as shown in Tables 3 and 5. (I do raise questions for this comparison in the question section.)

**Questions:**

I wonder how to fairly compare ATLAS and LLM since they may not use the same datasets.

(1) Saying ATLAS uses data A to perform pretraining and data B as a corpus, and LLM uses data C for pretraining, does A+B=C? If not, how to fairly compare them?

(2) The second question is also raised from the extra cost of using an external dataset in retrieval augmented generation (RAG). Though it's said in the paper ATLAS has fewer parameters 'despite having substantially fewer parameters', whether it's comparable to LLM measured on inference cost is not clear to me.

---

> ### Author Response · Authors · 2023-11-18
> **Response to Reviewer y5kL**
>
> Thank you for recognizing that our work is interesting, well-written, and that the performance improvement is significant. We appreciate your insightful feedback. Below, we address your comments and questions:
>
> > **Weakness 1: “Though the proposed method has a signiìcantly better few-shot performance than ATLAS…”**
>
> In the aspect of modeling, the core architecture of RAVEN builds upon ATLAS. This point also holds true for most LLMs, as they typically build upon similar architectures. However, it's crucial to emphasize that our methodologies weren't spontaneously conceived; they stem from an in-depth understanding of the limitations of current models and a careful consideration of how to overcome these issues as discussed in Sections 3 and 4. While the technique's design might appear simple, its conception was far from obvious without our analytical groundwork. Moreover, simplicity (when effective) is a commendable attribute of a technique.
>
> In summary, our work offers contributions from both analytical and technological perspectives:
> - From an analytical standpoint, we provide a thorough analysis of the in-context learning capability of state-of-the-art retrieval-augmented encoder-decoder language models. Our insights also highlight potential avenues for improvement. This analysis serves as the foundation for our proposed RAVEN training strategy and Fusion-in-Context prompting approach.
> - From a technological perspective, we introduce a confluence of retrieval-augmented masked and prefix language modeling, coupled with our Fusion-in-Context Prompting and In-Context Example Retrieval strategies. These techniques not only enhance the base model's capabilities (by mitigating the mismatch between pretraining and testing) but also maximize the potential of in-context learning for these models (either by scaling the number of in-context examples with FiCL or by retrieving better in-context examples).
>
>
>
> > **Weakness 2: Match Decoder-Only LLMs**:
>
> Our results indicate that RAVEN, despite having substantially fewer parameters, achieves comparable performance to much larger decoder-only models in certain scenarios. This is a notable achievement, considering RAVEN's smaller scale. We acknowledge that there is still a performance gap with larger models like PaLM 540B and Codex 175B in some cases, but given RAVEN's size and the training budget, its performance is competitive.
>
> > **Question 1: Fairness of Comparison**:
>
> This is an excellent point. Other LLMs are trained on much larger data, so we cannot expect a model trained on an academic budget to match the performance of these models. However, our model, despite having substantially fewer parameters, achieves comparable performance in certain scenarios. The comparison between RAVEN and ATLAS can be considered fair since they are trained on the same data.
>
> Furthermore, our aim is not to build the SOTA model, as this is impossible considering the budget, and no model stronger than GPT-4 truly exists yet. Our main goal is to demonstrate the possibility of in-context learning with retrieval-augmented encoder-decoder language models as a research practice. We provide comprehensive analysis and design techniques that enable these models to perform better in in-context learning.
>
> > **Question 2: Inference Cost and Efficiency**:
>
> This is a great point. While ATLAS is known for having fewer parameters, the incorporation of an external corpus for retrieval in RAG does contribute to the overall computational load. This is also true for RAG when used with decoder-only language models.
>
> For decoder-only LMs, the computational cost typically increases quadratically with the input length, as well as with the number of retrieval passages. In contrast, for encoder-decoder LMs with a fusion-in-decoder architecture, the computation cost grows linearly with the number of retrieved passages, as they only perform self attention over one passage at a time (Izacard et al., 2021). Therefore, from the perspective of computational overhead increasing with the number of retrieved passages, retrieval-augmented encoder-decoder language models are superior to decoder-only language models.

---

> ### Author Response · Authors · 2023-11-21
> **Thank you!**
>
> Dear Reviewer y5kL, we hope our response addresses your concerns. Please let us know your thoughts, and we are more than happy to answer any further questions.

---

> ### Comment · Reviewer_y5kL · 2023-11-22
>
> I thank the author for providing detailed responses to the weaknesses and questions from my review.
>
> Firstly, I think the analysis is good but not surprising. Then I still think the technique here is not really new. Prefix language modeling is an already known technique to improve performance, thus I won't count it much as a contribution (for instance I won't count data augmentation as a contribution especially if I propose some special data augmentation for my pipeline). Fusion-in-Context Prompting is new but it's kind of natural to the RAG pipeline, I would count it much for the contribution.
>
> Overall I think the technique contribution is weak. This is my major concern which leads to my slight negative score.

---

### Official Review · Reviewer_5UMJ · 2023-11-01

**Soundness:** 4 excellent
**Presentation:** 4 excellent
**Contribution:** 4 excellent
**Rating:** 6
**Confidence:** 3

**Summary:**

The paper focuses on advancements in language modeling, particularly exploring the training, performance, and applications of two versions of a model named RAVEN, in comparison with other models like ATLAS. Key aspects of the paper include:

1. Fusion-in-Context Learning: A notable contribution is the introduction of Fusion-in-Context Learning. This strategy is designed to address the limitations of constrained context length in base models like T5 or ATLAS, aiming to improve scalability and performance in in-context learning scenarios.

2. Comparative Performance Analysis: The paper presents a comparative analysis of RAVEN against other prominent models such as GPT-3, PaLM, Codex, and RETRO. This analysis, focusing on tasks like Natural Questions and TriviaQA, demonstrates that RAVEN, especially with FiCL, shows enhanced performance in both zero-shot and few-shot learning scenarios.

**Strengths:**

1. Innovative Approach: The introduction of Fusion-in-Context Learning (FiCL) to address the limitations of constrained context length in base models like T5 or ATLAS is a significant innovation. This strategy improves scalability and performance in in-context learning scenarios.

2. Comparative Performance: RAVEN models demonstrate superior performance compared to other models like GPT-3, PaLM, Codex, and RETRO in tasks such as Natural Questions and TriviaQA, particularly in zero-shot and few-shot settings.

**Weaknesses:**

1. The paper demonstrates the effectiveness of the proposed method primarily in the context of encoder-decoder models. However, its effectiveness in popular left-to-right language models, which are widely used, is not explicitly addressed. This omission can limit the understanding of how the proposed method might perform or be adapted to these prevalent LMs such as LLaMA and GPT.

**Questions:**

N/A

---

> ### Author Response · Authors · 2023-11-18
> **Response to Reviewer 5UMJ**
>
> Thank you for highlighting the innovation and significance of our methods, and for your insightful feedback.
>
> The Fusion-in-Context Learning (FiCL) strategy is a natural advantage of encoder-decoder language models. For decoder-only language models, FiCL may be adapted as follows: we can feed different in-context examples to the decoder and then ensemble the outputs.
>
> Another promising future direction is exploring how to combine the fusion-in-decoder architecture with the most powerful decoder-only language models. By doing so, we can harness the advantages of both architectures – employing a fusion-in bidirectional architecture to effectively encode retrieved passages for the most powerful decoder-only LLMs.

---

### Author Response · Authors · 2023-11-18
**General Response to Reviewers**

We thank all the reviewers for the insightful and constructive feedback. We appreciate the reviewers' positive comments about our work being innovative (5UMJ), interesting (y5kL, 2TC1), well-motivated (2TC1), well-written (y5kL, 2TC1), and effective (5UMJ, y5kL).

We have addressed each reviewer's concerns and questions in the individual responses below. Please let us know if you have any further questions or need clarifications. Thank you!